# Worst-Case-Optimal Joins on Graphs with Topological Relations

## ABSTRACT

Spatial data play an important role in many applications built over knowledge graphs, and are frequently referenced in queries posed to public query services, such as that of Wikidata. Querying for spatial data presents a significant challenge, as topological relations such as *adjacent* or *contains* imply inferred information, such as through the transitivity of the containment relation. However, despite all the recent advances in querying knowledge graphs, we still lack techniques specifically tailored for topological information. Applications looking to incorporate topological relations must either materialize the inferred relations, incurring high space and maintenance overheads, or query them with less efficient recursive algorithms, incurring high runtime overheads.

In this paper we address the problem of leveraging topological information in knowledge graphs by designing efficient algorithms to process these queries. Our solution involves building a specific index that stores the topological information in a convenient compact form, and includes specialized algorithms that infer every possible relation from the basic topological facts in the graph. We show that, while using essentially the same space required to solve standard graph pattern queries, we can incorporate topological predicates, accounting for all the inferred information, all within worst-case-optimal time. We implement our scheme and show experimentally that it outperforms baseline solutions by a notable margin.

ACM Reference Format:
Anonymous Author(s). 2025. Worst-Case-Optimal Joins on Graphs with Topological Relations. In *Proceedings of The Web Conference (TheWebConf '25)*. ACM, New York, NY, USA, 13 pages. https://doi.org/XXXXXXX.XXXXXXX

## 1 INTRODUCTION

Knowledge graphs are composed of diverse types of binary relations as distinguished by edge labels [27]. An individual relation may have specific semantic conditions, for example, transitivity, symmetry, asymmetry, antisymmetry, reflexivity, irreflexivity, etc. Though a wide variety of graph database engines [3, 38] have been proposed that can manage and query knowledge graphs, they typically do not take into account the diverse semantics of different relations. For example, a recent breakthrough for evaluating database queries more efficiently has been the development of *worst-case-optimal* (*wco*) join algorithms [43], which can help improve the performance of graph databases [1, 5, 10, 28, 37, 44, 53, 56]. However, such techniques are agnostic to the semantics of relations.

In the geospatial domain, the semantics of topological relations play a crucial role. For example, if region $A$ borders region $B$, region $A'$ contains $A$, region $B'$ contains $B$, and regions $A'$ and $B'$ do not overlap, this gives the implicit relation *region $A'$ borders region $B'$*. We focus on querying such topological relations, which incorporate spatial data by means of containment, disjointness, and adjacency relations, and which form a key part of a variety of open knowledge graphs on the Web, including DBpedia [33], LinkedGeoData [51], Wikidata [55], among various others. Other kinds of hierarchical information – such as the taxonomies present in DBpedia [33], Wikidata [55], etc. – exhibit topological semantics as well.

Though topological databases have been used in geographic information systems for decades [50], querying topology in knowledge graphs is not well-supported: topological databases based on the relational paradigm are not well-suited for querying knowledge graphs, while, to the best of our knowledge, no graph database engine supports topological relations directly. Though the semantics of such relations can be captured via *regular path queries (RPQs)* and relational algebra, the translation is cumbersome, and the performance of such rewritings in existing engines leaves (as we show later) much room for improvement. This highlights the need for efficient support for querying topological relations in graph databases.

In this paper, we study how to evaluate basic graph patterns (BGPs), as form the core of modern graph query languages [2], with topological relations. To illustrate, consider the topological relations contains$(x, y)$, disjoint$(x, y)$, and touches$(x, y)$ between regions. One may consider fully materializing these relations prior to querying. There are two problems with this approach, however.

Firstly, there is much *inferred* information that must either be explicitly encoded in the table, or obtained at query time with other mechanisms. But materializing fully inferred topological relations requires a prohibitive amount of space (an obvious case is the contains relation, which would require storing all transitive containment relations). On the other hand, using existing mechanisms such as RPQs results in complex queries involving combinations of RPQs and linear algebra that are hard to evaluate (see Section 8), and for which wco guarantees are not believed to exist [12, 15].

Secondly, querying negated version of the relations – like asking for non-contained or non-adjacent regions – requires either encoding (typically huge) tables for not-contains$(x, y)$, not-disjoint$(x, y)$, and not-touches$(x, y)$, or handling negations of complex queries in the language, and paying the corresponding price in performance.

We show how to efficiently handle such topological relations in both space and time. Concretely, take a graph database with $n$ nodes, $N$ labeled edges (or triples) and $M$ base adjacency relations between nodes (from which others can be derived). We can then index the graph using $O(n+N+M)$ space so that BGPs extended with containment and adjacency constraints, plus their negations, can be answered in wco time, exactly as if we had stored the explicit tables contains$(x, y)$, disjoint$(x, y)$, touches$(x, y)$, not-touches$(x, y)$, etc. including all information inferred from the base relations.

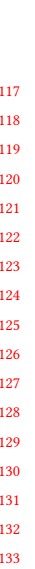

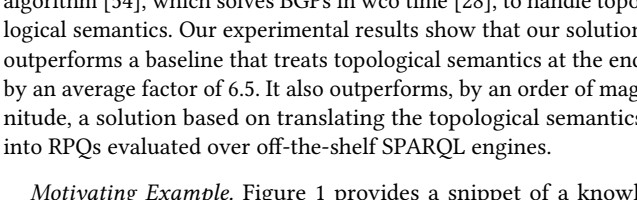

**Figure 1: Graph with two states in Northern and Central Africa, their topological relations and their languages**

To achieve this result we extend the Leapfrog Triejoin (LTJ) algorithm [54], which solves BGPs in wco time [28], to handle topological semantics. Our experimental results show that our solution outperforms a baseline that treats topological semantics at the end by an average factor of 6.5. It also outperforms, by an order of magnitude, a solution based on translating the topological semantics into RPQs evaluated over off-the-shelf SPARQL engines.

*Motivating Example.* Figure 1 provides a snippet of a knowledge graph describing regions, their topological relations (touches, disjoint and contains) and their official languages. Over this knowledge graph, we could consider posing the following query (BGP):

$$Q_1 = \{(\text{Africa}, \text{contains}, x), (\text{Africa}, \text{contains}, y),$$
$$(x, \text{touches}, y), (x, \text{language}, z), (y, \text{language}, z)\}$$

This query asks for pairs of regions in Africa that touch and share a language. We may expect this query to return:

| x | y | z |
|------|-------|--------|
| Chad | Libya | Arabic |
| Libya | Chad | Arabic |

However, if we run the query over the knowledge graph in a manner agnostic to the semantics of contains, we will receive empty results since the fact that Africa contains Chad and Libya (only) implicitly holds via the transitivity of this topological relation.

As another example, consider the simpler BGP:

$$Q_2 = \{(\text{Africa}, \text{contains}, x), (\text{Africa}, \text{contains}, y), (x, \text{touches}, y)\}$$

We may after some consideration expect the results to be:

| x | y |
|----------------|----------------|
| Chad | Libya |
| Libya | Chad |
| Central Africa | Northern Africa |
| Northern Africa | Central Africa |
| Central Africa | Libya |
| Libya | Central Africa |
| Northern Africa | Chad |
| Chad | Northern Africa |

Here, the fact that Central Africa and Northern Africa touch can be inferred from the observations that they are disjoint and contain two regions that touch. Likewise we can infer from the semantics of the indicated topological relations, and this graph, that Central Africa touches Libya, and Northern Africa touches Chad, with further results given by the symmetry of the touch relation.

The goal of this paper is to explore techniques for efficiently (in both space and time) evaluating BGPs with topological relations.

*Paper overview.* Section 2 recaps related works on spatial databases and topological relations in graphs. Section 3 defines the core concepts used throughout the paper. Section 4 discusses our extension of a seminal worst-case-optimal join algorithm to support topological relations. Section 5 outlines support for containment and disjointness relations, while Section 6 outlines support for adjacency relations. Section 7 proves the worst-case optimality of our scheme and then states our theoretical result. Section 8 discusses the implementation of our scheme and describes our experiments and results. Finally, Section 9 concludes.

## 2 RELATED WORKS

We now discuss works relating to spatial databases and topology in graph databases before highlighting novelty.

*Spatial databases.* Topological relations have long been studied in the context of geographical information systems (GIS). Egenhofer & Franzosa [18] present a seminal, set-based framework for defining sixteen topological spatial relations, nine of which are applicable for polygonal areas on a plane. The Region Connection Calculus (RCC) [48, 49], which was proposed around the same time, features eight binary topological relations for spatial regions.

Later works looked to integrate topological relations into spatial query languages and databases. Papadimitriou et al. [46] propose a query language based on eight of Egenhofer & Franzosa's topological properties: overlaps, disjoint, equal, meets (what we call touches), contains, covers (contains and shares a boundary), contained by, and covered by. They prove that the closure of these relations, in combination with typical logical operators, gives rise to a query language that is complete for topological queries.

Such relations are implemented in many of the spatial database systems used for GIS applications [50]. Modern predecessors of such systems include PostGIS [45], and spatial extensions of relational databases such as Oracle [29], Microsoft SQL Server [20], etc.

*Topology in graph databases.* A variety of open knowledge graphs on the Web contain topological relations, often from the geographic domain. Examples of knowledge graphs dedicated to geographic information include GeoNames[1], LinkedGeoData [51], WorldKG [17], etc. Other cross-domain open knowledge graphs, such as DBpedia [33], Wikidata [55], YAGO [26], etc., further contain topological relations capturing geographical and taxonomic information.

Query languages have emerged for querying graphs with spatial information, including GeoSPARQL [9], which, alongside support for geometry and distance, features three families of topological relations, including that of Egenhofer et al., and RCC. Implementations of GeoSPARQL include Parliament [9], with extensions also available in well-known engines, such as Jena[2]. A similar language, called stSPARQL [31], supports spatial and time features, including topological relations, as implemented by Strabon [32]. Other popular commercial graph database engines implement custom spatial features, including Neo4j[3], TigerGraph[4], etc.

Key applications involve a combination of networks/graphs in a spatial context, including topological analyses of power grids [30,

---

[1]See https://www.geonames.org/
[2]See https://jena.apache.org/documentation/geosparql/
[3]See https://neo4j-contrib.github.io/spatial/
[4]See https://www.tigergraph.com/solutions/geospatial-analysis/

47], road networks [21], genomes [16], as well as querying the geographic knowledge graphs previously mentioned [25].

*Novelty.* We address the efficient evaluation of basic graph patterns over graphs that feature topological (and non-topological) relations. To the best of our knowledge, this problem has not been well-studied, where the works described in this section would benefit from such techniques. We provide the first worst-case-optimal algorithm for evaluating BGPs with key topological relations, while also carefully addressing efficiency in terms of space and time.

## 3 CORE CONCEPTS

We introduce key concepts and notation that will be used throughout relating to graph databases, worst-case optimality, Leapfrog Triejoin, compact data structures, and topological relations.

### 3.1 Graph Databases

A graph database is defined herein as a labeled directed graph $G$, i.e., a set of edges of the form $s \xrightarrow{p} o$, from node $s$ to node $o$ with label $p$. Such edges are denoted by *triples* $(s, p, o) \in \mathcal{U}^3$, where $\mathcal{U}$ is a totally ordered universe of constants. We call $N = |G|$ the number of triples in $G$, and dom($G$) the *domain* of $G$, that is, the subset of $\mathcal{U}$ used as constants in $G$. For simplicity we assume dom($G$) = $\{1, \ldots, n\}$.

We query graph databases by means of *basic graph patterns* (BGPs). Let $\mathcal{V}$ be a universe of *variables*. A BGP $Q$ is a set of *triple patterns*, each of which is a tuple of the form $(s, p, o) \in (\mathcal{U} \cup \mathcal{V})^3$, that is, it combines constants from $\mathcal{U}$ and variables from $\mathcal{V}$. Let vars($Q$) denote the set of variables used in $Q$. A *binding* is a mapping from variables in vars($Q$) to constants in $\mathcal{U}$. Given $Q$ and $G$, we call a binding a *solution* if and only if binding the variables of $Q$ accordingly results in a subgraph of $G$. The problem is to output the set of all the solutions, denoted $Q(G)$.

### 3.2 Topological Relations

Our topological model consists of a set of objects, and the following binary relations between some pairs of objects:

- $x \sqsubseteq y$: $x$ is contained in $y$.
- $x \not\sqcap y$: $x$ and $y$ are disjoint.
- $x \mid y$: $x$ and $y$ are adjacent, or "touch".

We further consider the negated relations $x \not\sqsubseteq y$, $x \sqcap y$, $x \nmid y$: not contained in, not disjoint and not adjacent, respectively.

The relation $\sqsubseteq$ forms a *hierarchy*. A hierarchy is an order (i.e., it is reflexive, antisymmetric, and transitive) such that if $x \sqsubseteq y$ and $x \sqsubseteq z$, then $y \sqsubseteq z$ or $z \sqsubseteq y$. In other words, the Hasse diagram of $\sqsubseteq$ is a forest. We do not allow partial overlaps between objects: two regions are either disjoint or one contains the other. Formally, $x \sqcap y \Leftrightarrow x \sqsubseteq y \vee y \sqsubseteq x$ (so $x \sqcap y$ is symmetric). The relation $\mid$ is also symmetric. Two adjacent nodes are considered to be disjoint (i.e., nodes containing each other are not adjacent), per rule a1 next. Rule a2 implies that, if two nodes are adjacent, then all their ancestors are also pairwise adjacent unless one contains the other.

- a1. If $x \sqsubseteq y$ then $x \nmid y$; if $x \mid y$ then $x \not\sqcap y$.
- a2. If $x \mid y$, $x \sqsubseteq x'$, and $y \sqsubseteq y'$, then $x' \mid y'$, or $x' \sqsubseteq y'$, or $y' \sqsubseteq x'$.

Our setup is a specialization of Region Connection Calculus (RCC) [48, 49], which further allows objects to overlap. Figure 2

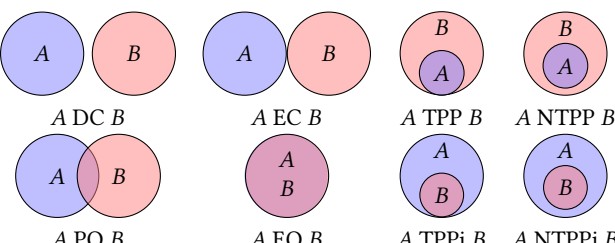

Figure 2: The 8 relations of the Region Connection Calculus

shows the 8 RCC relations, which are jointly exhaustive and pairwise disjoint, that is, every pair of objects has exactly one relation. Our relation $x \sqsubseteq y$ corresponds to '$x$ EQ $y$' or '$x$ TPP $y$' or '$x$ NTPP $y$' (or '$y$ TPPi $x$' or '$y$ NTPPi $x$'), without distinction. Relation $x \mid y$ corresponds to '$x$ EC $y$', and $x \not\sqcap y$ to '$x$ EC $y$' or '$x$ DC $y$'. Relation '$x$ EQ $y$' corresponds to $x = y$, and our model forbids '$x$ PO $y$'.

### 3.3 Worst-Case Optimality

The AGM bound [7] defines a limit on the number of solutions for natural join queries in a relational setting. Given a natural join query $Q$ and a relational instance $D$, the AGM bound of $Q$ over $D$ is the maximum number of tuples (results) generated by evaluating $Q$ over any instance $D'$ of size not greater than $D$. If we simply assume that the size of all relations is in $O(N)$, we can speak of the AGM bound of $Q$, denoted herein by $Q^*$, as a function of $N$. The AGM bound can also be extended to BGPs on graph databases [28], where $Q^*$ is the maximum number of solutions that BGP $Q$ may have on any graph database with $O(N)$ triples.

An algorithm finding all the solutions of a BGP $Q$ is said to be *worst-case optimal (wco)* if it runs in time $O(Q^*)$ in *data complexity* (i.e., assuming that the number of terms $|Q|$ in $Q$ is a constant). This is because, in the worst case, the algorithm has to enumerate $Q^*$ solutions, which requires $\Omega(Q^*)$ time. A logarithmic overhead factor (i.e., $O(Q^* \log N)$) is often permitted in wco algorithms to allow more flexibility in the underlying implementations. This is the case for Leapfrog Triejoin (LTJ), one of the most popular wco algorithms for BGPs on graph databases [28, 54].

### 3.4 Leapfrog Triejoin (LTJ)

The variant of the LTJ algorithm that solves BGPs in graphs proceeds by "eliminating" one variable at a time [28, 54], i.e. finding all candidate bindings for the variable that may lead to a solution.

LTJ first defines an initial ordering $(x_1, \ldots, x_v)$ of vars($Q$). Starting with $x_1$, LTJ finds each binding $c \in$ dom($G$) for $x_1$ such that, for every triple pattern $t$ where $x_1$ appears, if $x_1$ is replaced by $c$ in $t$, then the evaluation of the modified $t$ over $G$ is non-empty. This is equivalent to intersecting the binding of $x_1$ over all the individual triple patterns $t$ where $x_1$ appears. LTJ uses a procedure called 'seek' to find each consecutive value $c$ in that intersection. Procedure seek uses in turn the primitive leap($x_1, c$) to iteratively find, in each triple pattern $t$ where $x_1$ appears, the next possible candidate for the intersection, which corresponds to the smallest binding for $x_1$ in $t$ that is over some threshold $c$. When procedure seek finally finds a value $c$ that appears in all triple patterns $t$ where

$x_1$ appears, LTJ binds $x_1$ to $c$ and recursively continues eliminating the other variables. In each recursion branch where all $v$ variables have been eliminated, LTJ reports a *solution* with the bindings of $(x_1, \ldots, x_v)$. When the recursion returns to $x_1$, LTJ keeps finding new bindings for it from $c + 1$ onwards, and returns to the caller when all the bindings have been explored.

For LTJ to run in $O(Q^* \log N)$ time (i.e., for the algorithm to be wco), it suffices that primitive leap is supported in $O(\log N)$ time.

The Ring [4, 5] is a recent implementation of LTJ that retains its time complexity while requiring only $3N + o(N)$ words of space; note that $3N$ words is the space needed to just list the triples of $G$ in plain form. We build on this compact representation because we aim to extend LTJ while keeping the total space usage low.

## 3.5 Compact Data Structures

We make use of various compact data structures [41] in order to keep the space usage low. We present them in the depth that is needed to follow the paper.

*Bitvectors.* A bitvector is a sequence $B[1 \ldots n]$ that supports the following operations:

- Accessing the bit at position $i$, that is, $B[i]$.
- Computing the number $rank_b(B, i)$ of times bit $b \in \{0, 1\}$ occurs in the prefix $B[1 \ldots i]$.
- Finding the position $select_b(B, j)$ of the $j$th occurrence of the bit $b \in \{0, 1\}$ in $B$.
- Finding the position $succ_b(B, i)$ of the first occurrence of bit $b \in \{0, 1\}$ in $B[i \ldots n]$. This is indeed $select_b(rank_b(B, i-1) + 1)$, but it can be implemented more efficiently in practice.

All of these operations can be computed in $O(1)$ time by spending just $o(n)$ additional bits on top of $B$ [13, 39].

*Permutations.* A permutation $\pi$ on $[n]$ can be stored using $n \log n + O(\epsilon n)$ bits of space (our logarithms are in base 2) so that one can compute any $\pi(i)$ in time $O(1)$, and any $\pi^{-1}(j)$ in time $O(1/\epsilon)$ [40]. In this paper we use the setting $\epsilon = 1/\log n$, to use $n \log n + O(n)$ bits and accessing the inverse permutation in time $O(\log n)$.

*Ordinal trees.* An ordinal tree of $n$ nodes can be represented using a sequence of $2n$ parentheses: traverse the tree in DFS order, appending an opening parenthesis when arriving at a new node and a closing one after having visited all of its descendants. Those parentheses can then be represented as a bitvector $P[1 \ldots 2n]$ (say, encoding the opening parenthesis as a 1 and the closing one as a 0). By using $o(n)$ additional bits, one can carry out the following operations in $O(1)$ time, among many others [42]:

- The position $close(x)$ of the parenthesis that closes the one that opens at $P[x] = 1$.
- The position $open(x)$ of the parenthesis that opens the one that closes at $P[x] = 0$.
- The position $enclose(x)$ of the opening parenthesis that most tightly encloses that one at $P[x] = 1$.
- The position $rmq(x, y)$ where the minimum *excess* occurs between positions $x$ and $y$. The excess at position $z$ is the number of opening and not yet closed parentheses up to position $z$, that is, $rank_1(P, z) - rank_0(P, z)$.

*Wavelet matrices.* A wavelet matrix [14] represents an $n \times n$ grid with $M$ points using $M \log n + o(M \log n) + O(n)$ bits, while supporting various orthogonal range queries [8]. The queries we are interested in are finding the leftmost/rightmost/highest/lowest point in an orthogonal range of the grid. Those correspond to the operations called rel_min_lab_maj and rel_min_obj_maj, and analogous versions to find maxima instead of minima, on the structure BINREL-WT [8], which solves them in time $O(\log n)$.

## 4 QUERYING GRAPHS WITH TOPOLOGICAL RELATIONS

### 4.1 Model

We assume that (some) nodes of $G$ feature topological relations between them, per Section 3.2. These relations are encoded in the graph itself via triples of the form $(x, \text{contains}, y)$, $(x, \text{contained}, y)$, and $(x, \text{touches}, y)$. From those triples, which we call *axioms*, we derive the relations of Section 3.2 as follows.

- $x \sqsubseteq y$ iff axioms $(x, \text{contained}, y)$ or $(y, \text{contains}, x)$ occur in the graph, or it can be derived from axioms by reflexivity and transitivity. We use a closed-world assumption, so $x \not\sqsubseteq y$ iff we cannot derive $x \sqsubseteq y$.
- $x \mid y$ iff there is an axiom $(x, \text{touches}, y)$ or $(y, \text{touches}, x)$ in the graph, or it can be derived from rule a2 of Section 3.2. Again, $x \nmid y$ holds iff we cannot derive $x \mid y$.
- $x \sqcap y$ iff we can derive $x \sqsubseteq y$ or $y \sqsubseteq x$, otherwise $x \sqcap\!\!\!/\, y$ holds.

Note that rules like a1 or the antisymmetry of $\sqsubseteq$ are not used to derive relations; the axioms must hold them for consistency.

We then extend our BGPs with additional *constraints* of the form $x \sqsubseteq y$, $x \sqcap y$, $x \mid y$, and their negations, where $x$ and $y$ are either constants or variables. The solutions to these *extended BGPs* are the bindings such that, once the variables are substituted, all the triples appear in the graph and all topological constraints (per the above itemization) are satisfied.

### 4.2 Algorithms

To achieve the promised optimality, we extend LTJ so that it can handle the constraints while maintaining the leap$(x, c)$ operation working in time at most $O(\log n)$. The treatment of the constraints differs depending whether $x$ or $y$ is bound first.

- For $x \sqsubseteq y$, if $x$ is bound, leap$(x, c)$ is implemented with function contained$(x, c)$, which returns the smallest $y \geq c$ such that $x \sqsubseteq y$. If $y$ is bound, then leap$(y, c)$ is implemented with function contains$(y, c)$, which returns the smallest $x \geq c$ such that $x \sqsubseteq y$. The negated versions, for $x \not\sqsubseteq y$, are implemented with functions not-contained$(x, c)$ and not-contains$(y, c)$, respectively.
- For $x \sqcap\!\!\!/\, y$, if $x$ is bound, leap$(x, c)$ is implemented with function disjoint$(x, c)$, which returns the smallest $y \geq c$ such that $x \sqcap\!\!\!/\, y$. If $y$ is bound, leap$(y, c)$ is similarly implemented with disjoint$(y, c)$. The negated version, for $x \sqcap y$, is implemented with function not-disjoint$(x, c)$ (or not-disjoint$(y, c)$).
- For $x \mid y$, if $x$ is bound, leap$(x, c)$ is implemented with function touches$(x, c)$, which returns the smallest $y \geq c$ such that $x \mid y$; the case of bound $y$ uses touches$(y, c)$. The negated

version, for $x \nmid y$, is implemented with function not-touches $(x, c)$ (or not-touches$(y, c)$).

If both $x$ and $y$ are constants, we just check if the fact holds and, if it does, we remove it from the extended BGP; otherwise the query has no results. The remaining case is that both $x$ and $y$ are variables; to handle it within LTJ we also create functions contained$(c)$, contains$(c)$, etc., which return the smallest $z \geq c$ that contains some node, is contained in some node, and so on.

In Sections 5 and 6 we describe how we handle those functions efficiently. Some of them are implemented in constant time and others in time $O(\log n)$; the only exception is not-touches, which takes time $O((h/\epsilon) \log n)$ where $h$ is the height of the hierarchy and $\epsilon$ is a space-time tradeoff parameter. This yields our main result; see the details in Appendix A.1.

**Theorem 1.** *Let $G$ be a graph with $n$ nodes and $N$ triples. We can build an index of size $O(n + N)$ that solves any extended BGP $Q$ not involving constraints of the form $x \nmid y$ with exactly one variable, in time $O(Q^* \cdot |Q| \log n)$, where $Q^*$ is the maximum size of the output of $Q$ on any graph with at most $|Q|n$ nodes and $|Q|N$ triples. Given a query without such constraints, we can choose a parameter $1 \leq \epsilon \leq h$ at index construction time so that the index uses $O(\epsilon N)$ additional space and time becomes $O(Q^* \cdot |Q|(h/\epsilon) \log n)$, for $h$ the maximum length of a chain of containment relations between nodes of $G$.*

## 5 CONTAINMENT AND DISJOINTNESS

The main idea of our data structure handling containment and disjointness relations is to store the Hasse diagram of the relation $\sqsubseteq$ (which is a forest) using balanced parenthesis as shown in Section 3.5. However, we also renumber the node identifiers, which form an interval $[1 \mathinner{.\,.} n]$, so as to assign their *postorder* number in this forest (see, e.g., [34, 57]). Recall that a postorder visits first the children of a node, left to right, and then visits the node. The postorder numbers will be the identifiers used internally for indexing and querying. The mapping with the external identifiers, if necessary, will be provided with a permutation $\pi : [1 \mathinner{.\,.} n] \to [1 \mathinner{.\,.} n]$, so that $\pi(i)$ will be the external identifier of the node with postorder number $i$. Function $\pi$ will be stored as described in Section 3.5, so that the translation of query results takes constant time, while the external identifiers appearing in BGPs $Q$ can be translated into their corresponding internal identifiers (i.e., postorder numbers) in time $O(|Q| \log n)$ using the operation $\pi^{-1}(j)$.

As we explain in Section 3.5, we can represent the Hasse diagram of $\sqsubseteq$ in $2n + o(n)$ bits using balanced parentheses. If there are several trees in the forest, we concatenate their parenthetical representations. Since we use the postorder numbering of nodes, in this paper the identifier of a tree node will be the position of its *closing* parenthesis. Our representation supports the following primitives (among others) in constant time:

- $node(i) = select_0(i)$ gives the forest node with postorder $i$,
- $postorder(x) = rank_0(x)$ yields the postorder of node $x$,
- $first(x) = 1 + rank_0(open(x))$ gives the least postorder in the subtree rooted at $x$,
- $lca(x, y) = close(enclose(rmq(x, y) + 1))$ gives the lowest common ancestor of nodes $x$ and $y$.

Our postorder numbering has convenient properties:

(1) All the nodes contained in $x$ form a range of postorders, that is $x \sqsubseteq y$ iff $first(y) \leq postorder(x) \leq postorder(y)$;

(2) Let us call $ancestor(x, j)$ the $j$th ancestor of node $x$ (where $j = 0$ yields $x$ itself and $j = 1$ gives its parent). Then the sequence $postorder(ancestor(x, j))$ is increasing with $j$.

We can then implement the containment operations as follows, all in constant time. Figure 3 illustrates the operations.

contains$(x, c)$: Return $c$ if $node(c) \sqsubseteq x$, as $node(c)$ is already in the subtree of $x$. Otherwise, if $c < postorder(x)$ return $first(x)$, the first postorder following $c$ that is below $x$. Else return $\perp$, as $c > postorder(x)$ and no postorder following $c$ can be inside $x$.

not-contains$(x, c)$: Return $postorder(x) + 1$ if $node(c) \sqsubseteq x$, as that is the least postorder following $c$ of a node not contained in $x$. Otherwise return $c$, as $node(c)$ is already out of the subtree of $x$.

contained$(x, c)$: Return $x$ if $c \leq postorder(x)$, because $x$ has the least postorder among its ancestors. Otherwise, $c$ is on, or departs from, the root-to-$x$ path at a node $y = lca(x, node(c))$. The answer is then $postorder(y)$, as we prove next (see Appendix A.2).

**Lemma 1.** *Let $x$ and $z$ be such that $postorder(x) < postorder(z)$. Then $y = lca(x, z)$ is the node with the minimum $postorder(y) \geq postorder(z)$ that is an ancestor of $x$.*

not-contained$(x, c)$: Return $c$ if $x \not\sqsubseteq node(c)$, as $c$ is already not an ancestor of $x$. Otherwise, we should climb the path formed by $ancestor(node(c), j)$, $j = 1, 2, \ldots$ until finding an ancestor $y$ having another child to the right, and the answer is the first postorder under $y$. This is easily detected in the parentheses representation $P[1 \mathinner{.\,.} 2n]$ of the tree: we want to find the first opening parenthesis to the right of $P[node(c)]$, which is done in constant time with $j = succ_1(P, node(c)) - node(c)$ (cf. Section 3.5); the answer is $c + j$.

*Disjointness.* Based on the containment algorithms, we answer disjoint$(x, c)$ by returning $c$ if $c < first(x)$. Otherwise, if $c \leq postorder(x)$, then $node(c) \sqsubseteq x$, so we reset $c \leftarrow postorder(x) + 1$ to get out of the area below $x$. Finally, once we have ensured that $c > postorder(x)$, we simply return not-contained$(x, c)$. To answer not-disjoint$(x, c)$, we return $first(x)$ if $c < first(x)$. Otherwise, we return $c$ if $node(c) \sqsubseteq x$. Otherwise, $c > postorder(c)$ and we just return contained$(x, c)$. In all cases the process takes constant time.

*Zero or two bound variables.* See Appendix A.3.

## 6 ADJACENCY CONSTRAINTS

We assume $M$ adjacency axioms of the form $(x, \text{touches}, y)$ in $G$, from which the whole set of adjacency relations are derived.

We use a data structure that takes $M \log n(1 + o(1))$ bits of space and infers all the derived relations, answering the queries in time $O(\log n)$. The data structure uses an $n \times n$ binary matrix $A$ containing $2M$ 1s, where the rest are 0s: for each axiom $(x, \text{touches}, y)$ or $(y, \text{touches}, x)$, we set 1s at $A[postorder(x), postorder(y)]$ and $A[postorder(y), postorder(x)]$ (we later tighten this space).

Recall that the set of postorders of all the descendants of $x$ forms a range $[first(x) \mathinner{.\,.} postorder(x)]$. By rule a2, every $A[i, j] = 1$ where $first(x) \leq i \leq postorder(x)$ implies that $x \mid node(j)$, because some descendant of $x$ is adjacent to $node(j)$—unless $x$ contains $node(j)$.

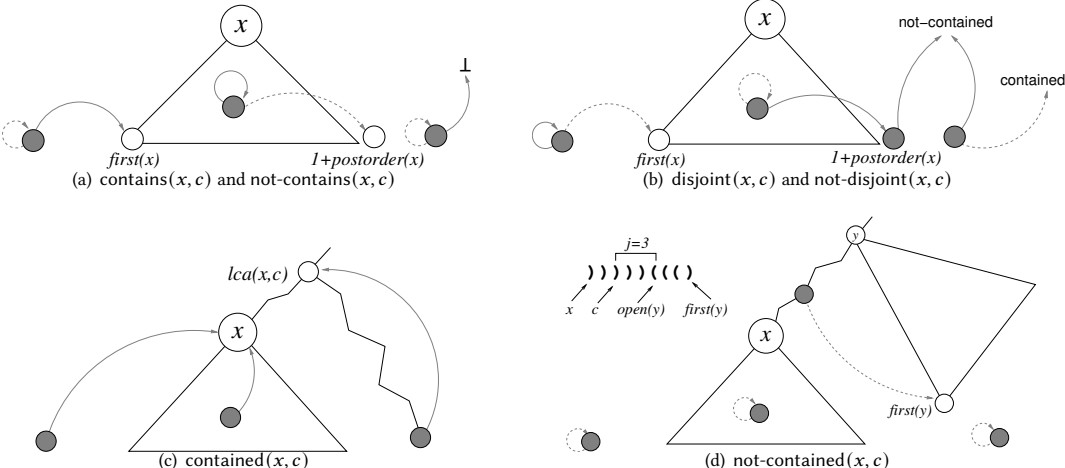

**Figure 3: Illustration of operations (solid curves) and their negations (dashed curves), for all cases of $c$ (grayed node). The parentheses on the top left of Figure 3(d) illustrate how the distance $j$ between $c$ and the answer postorder is found.**

Furthermore, every ancestor of $node(j)$ is also adjacent to $x$, again by rule a2—unless that ancestor of $node(j)$ contains $x$.

To compute touches$(x, c)$, we want the least postorder $\geq c$ of a node that is adjacent to $x$. We consider two ranges of columns in $A$.

*Columns $j \geq c$.* Each $A[i, j] = 1$ with $first(x) \leq i \leq postorder(x)$ and $j \geq c$ implies $x \mid node(j)$, and thus $j$ can be the answer, unless $x$ and $node(j)$ contain one another. By rule a1, $x \not\sqsubseteq node(j)$, as otherwise $node(i) \sqsubseteq node(j)$ holds. Yet, it may be that $node(j) \sqsubseteq x$.

We handle the case $j \geq c$ as follows. Let $j$ be minimal such that $A[i, j] = 1$ for some $first(x) \leq i \leq postorder(x)$ and $j \geq c$. If $node(j) \not\sqsubseteq x$, then $j$ is the best answer from the matrix columns $[c \ldots n]$: the ancestors of $node(j)$ have larger postorders, and other 1s in this area of $A$ have larger postorders, too. If $node(j) \sqsubseteq x$, then the least $j \geq c$ is below $x$, and therefore we must find the leftmost answer in the columns $[postorder(x) + 1 \ldots n]$, that is, we set $c \leftarrow postorder(x) + 1$ and find again the minimal $j$. This second time, if there is a new answer $j$, it cannot be contained in $x$.

*Columns $j < c$.* For each $A[i, j]$ with $first(x) \leq i \leq postorder(x)$ and $j < c$, an ancestor of $node(j)$ can be the answer, unless $x \sqsubseteq node(j)$ or $node(j) \sqsubseteq x$. The first case cannot occur by rule a1. If the second case occurs, then the ancestors of $node(j)$ are not suitable either, as they contain or are contained in $x$.

We handle the case $j < c$ as follows. Let $j$ be maximal such that $A[i, j] = 1$ for some $first(x) \leq i \leq postorder(x)$ and $j < c$. If $node(j) \not\sqsubseteq x$, we use Lemma 1 to find in $y = lca(node(j), node(c))$ the ancestor of $node(j)$ with minimum $postorder(y) \geq c$; note that no ancestor of $j$ can be contained in $x$. If instead $node(j) \sqsubseteq x$, then the rightmost $j < c$ is below $x$ and we must find the maximum $j$ in columns $[1 \ldots first(x) - 1]$ of $A$, that is, we set $c \leftarrow first(x) - 1$ and find the maximal $j$ and compute $y$ again. This second time $j$ will not be contained in $x$; still we must discard $y$ if it contains $x$.

The reason why we can just take Lemma 1 on the rightmost suitable $j$ is given next (see Appendix A.4). It shows that using Lemma 1 on $y$ yields a smaller answer than using $x$.

**Lemma 2.** *Let $x, y,$ and $z$ be such that $postorder(x) < postorder(y) < postorder(z)$. Then $postorder(lca(x, z)) \geq postorder(lca(y, z))$.*

*The actual algorithm.* We have shown that, if we store each adjacency axiom twice, then we need to find the leftmost/rightmost 1 in a 3-sided area of $A$ $O(1)$ times to get the best candidate in columns $[c \ldots n]$ and $[1 \ldots c - 1]$, and then we can pick the smallest of two answers. To store each axiom $(x, touches, y)/(y, touches, x)$ only once, we arbitrarily store $A[postorder(x), postorder(y)] = 1$ or $A[postorder(y), postorder(x)] = 1$. The answer to touches$(x, j)$ is then found as the minimum of four candidates: two obtained as described, and other two with the analogous query in the other direction (i.e., on 3-sided areas $A[1 \ldots c-1][first(x) \ldots postorder(x)]$ and $A[c \ldots n][first(x) \ldots postorder(x)]$).

To solve those orthogonal range queries within the promised space, we resort to the wavelet matrices described in Section 3.5, which use $M \log n + o(M \log n) + O(n)$ bits and carry out those queries in time $O(\log n)$. This is also the complexity of touches$(x, c)$.

*Negated adjacency.* A basic solution for not-touches$(x, c)$ is to invoke touches$(x, c+j)$ for $j = 0, 1, \ldots$ until touches$(x, c+j) > c+j$, so we can answer not-touches$(x, c) = c+j$. By using $O(\epsilon M)$ further words of space, for any $\epsilon \leq h$, we can guarantee a time bound of the form $O((h/\epsilon) \log n)$, where $h$ is the maximum height of a tree in the Hasse diagram of the relation $\sqsubseteq$. For every node $x$, we consider all the nodes $y$ such that $x \mid y$ (not only as an axiom, but as a derived fact as well). We then store a list, associated with $x$, of all the maximal *long runs* of $\lceil h/\epsilon \rceil$ or more consecutive values in this set.

Since there are at most $hM$ derived adjacency relations, there can be at most $\epsilon M$ runs to store, which yields the promised space. On the other hand, we can try the basic method for $j = 0, 1, \ldots, h/\epsilon$, and if we fail for them all, then we are inside a long run, which can be searched in the list in logarithmic time so as to return the first value following the run. We can choose, for example, a constant $\epsilon$ to have $O(M)$ space and $O(h \log n)$ time, or some $\epsilon = \Theta(h)$ to have $O(hM)$ space and $O(\log n)$ time.

*Zero or two bound variables.* See Appendix A.5.

## 7 WORST-CASE OPTIMALITY

In this section we show that the extended LTJ is worst-case optimal: its running time, for an extended BGP $Q$ and a graph $G$ is always bounded by the maximum number of answers of $Q$ over any graph $G'$ with (about) the same number of nodes and edges.

To prove worst-case optimality, we first bound the number of answers of an extended BGP, and then show the extended LTJ algorithm runs in time given by this bound. We only focus on extended BGPs $Q$ that are *consistent*, that is, if there is at least one graph $G$ for which $Q(G)$ is nonempty. This condition can be checked in polynomial time [24].

*Bounding the number of answers.* For an extended BGP $Q$, consider the conjunctive query $flat(Q)$ that has an atom $T_{y,z,w}(y,z,w)$ for each pattern $(y,z,w) \in Q$, and atoms $D_C(u), D_C(v)$ for each constraint $C$ in $Q$ that mentions variables $u$ and $v$. Using the techniques introduced by Cucumides et al. [15], it is not difficult to show that the size of $Q(G)$, for a graph $G$ with $N$ triples and $n$ nodes, is upper bounded, in data complexity, by the size of the evaluation of $flat(Q)$ over the database instance $I(G)$ in which each $T_{x,y,z}$ contains every triple in $G$, and $D_C$ contains every node[5]. Hence, $Q(G)$ is always bounded by $2^{\rho^*(flat(Q),(N,n))}$, where $\rho^*(flat(Q),(N,n))$ is the AGM bound of $flat(Q)$ over instances with $N$ triples and $n$ nodes [7]. Next we show a matching lower bound. Unlike the upper bound, this result does not follow from the techniques of Cucumides et al.[15], as it is deeply related to the topological constraints in graphs (see Appendix A.6). Furthermore, notice the result is slightly weaker than the original AGM bound. This is due to the presence of self joins in extended BGPs (see e.g. [23]).

PROPOSITION 2. *Given an extended BGP $Q$ with $\ell$ triple patterns, there are arbitrarily large graphs $G$ with $\ell N$ triples and $\ell n$ nodes for which $Q(G) \geq \rho^*(flat(Q),(N,n))$.*

*Analyzing the algorithm.* Next, we show our algorithm does indeed run in worst-case-optimal time. In the following we use $Q^*$ to refer to $2^{\rho^*(flat(Q),(N,n))}$; in view of Proposition 2, we can assert that $Q^*$ is the maximum size of the output of $Q$ over any graph with at most $|Q|N$ triples and $|Q|n$ nodes.

PROPOSITION 3. *The extended LTJ algorithm runs on an extended BGP $Q$ over a graph $G$ in time $O(Q^* \cdot |Q| \log n)$ if $Q$ does not use the not-touches constraint, and in $O(Q^* \cdot |Q|(h/\epsilon) \log n)$ for queries involving the not-touches constraint, where $h$ is the maximum length of a chain of containment relations between nodes of $G$ and $1 \leq \epsilon \leq h$.*

PROOF. Recall that standard LTJ runs in time $O(Q^* \cdot |Q| f(N,n))$ when leap operations are implemented in time $O(f(N,n))$ [54].

Consider any ordering of variables. We show that the running time of our algorithm is bounded by the running time taken by the Ring [4] to process $flat(Q)$ over $G$. Budget allocation is as follows. As long as we do not bind any variable participating in a topological constraint, budget is allocated directly as Ring operations are identical. Now whenever we bind a variable $x$ that participates

---

[5]Note that this upper bound does involve a constant that depends on the query, the reason is that the original AGM bound is given for join queries, which do not repeat relations, while our queries do repeat relations. See [15] for further discussion.

in a constraint, the number of leap operations for $x$ we do when processing $Q$ is at most the number of leaps for processing $flat(Q)$: any constraint $C(x,y)$ is replaced in $flat(Q)$ with $D(x), D(y)$, so constraints can only reduce the number of leap operations.

The Ring implements the leap operations in time $O(\log n)$ [4]. In the preceding sections we have shown that leap operations on topological constraints can be performed in time $O(\log n)$, or $O((h/\epsilon) \log n)$ if the constraint is of type not-touches. This gives our desired running time bound. □

## 8 IMPLEMENTATION AND EXPERIMENTS

We implemented our index in C++ as an extension of the Ring [4, 5], which solves basic BGPs using LTJ in little space. Like the Ring, our implementation is single-threaded and it is built on top of the Succinct Data Structures Library (SDSL) [22]. Our implementation is available at https://anonymous.4open.science/r/Toporing-6FD5/.

To process extended BGPs, we implemented the leap procedures exactly as described in Sections 5 and 6. The LTJ algorithm is then applied without changes, other than detecting the topological triple patterns so as to treat them in special form. Our index is called *TopoRing* in the experiments. More details are given in Appendix B.

*Dataset.* To test our solution we queried the truthy Wikidata graph [55] with 2,762 real-world graph patterns extracted from the Wikidata query logs [36]. Further details are given in Appendix B.

*A baseline.* For comparison, we developed a non-trivial (but not worst-case optimal) baseline based on the Ring and the data structures presented in Sections 5 and 6. Unlike our solution, where both standard and topological triple patterns in a BGP are processed together following the LTJ process, the baseline works in two steps: *i)* First, process all the standard triple patterns of the BGP using the Ring in order to obtain a partial binding of values to variables, and then *ii)* filter or extend each partial binding with the topological triple patterns, using the ideas in Sections 5 and 6. In part *ii)*, the triples with two bound variables are processed first, as they serve to filter the solutions. We then continue with triples with one bound variable, which are used to extend the binding. Finally, we end with triples with both unbound variables. Notice that when a variable is bound during evaluation, other triples having that variable reach a higher priority to be processed next.

*Virtuoso, Blazegraph and Jena.* Our topological queries can be expressed in SPARQL, albeit resorting to more complex queries that combine BGPs, RPQs and negation. Along these lines we also include the Virtuoso [19], Blazegraph [52] and Apache Jena [11] SPARQL engines, translating queries into their equivalent SPARQL syntax; Appendix B gives the details. To provide a fairer comparison with our in-memory solution, we tested the engines on a RAM disk.

### 8.1 Experimental results

*Topological primitives.* We measured the standalone time of our primitives contains$(x,c)$, contained$(x,c)$, touches$(x,c)$, and their negations not-contains$(x,c)$, not-contained$(x,c)$ and not-touches $(x,c)$ (with $\epsilon = 2$), obtaining 9.5, 14.1, 27.3, 9.4, 9.0, and 30.6 nanoseconds, respectively. Those times are the average of 20 million queries over random subjects $x$ and random valid objects $c$. A standard leap$(x,c)$ on the Ring was much slower, 340 nanoseconds.

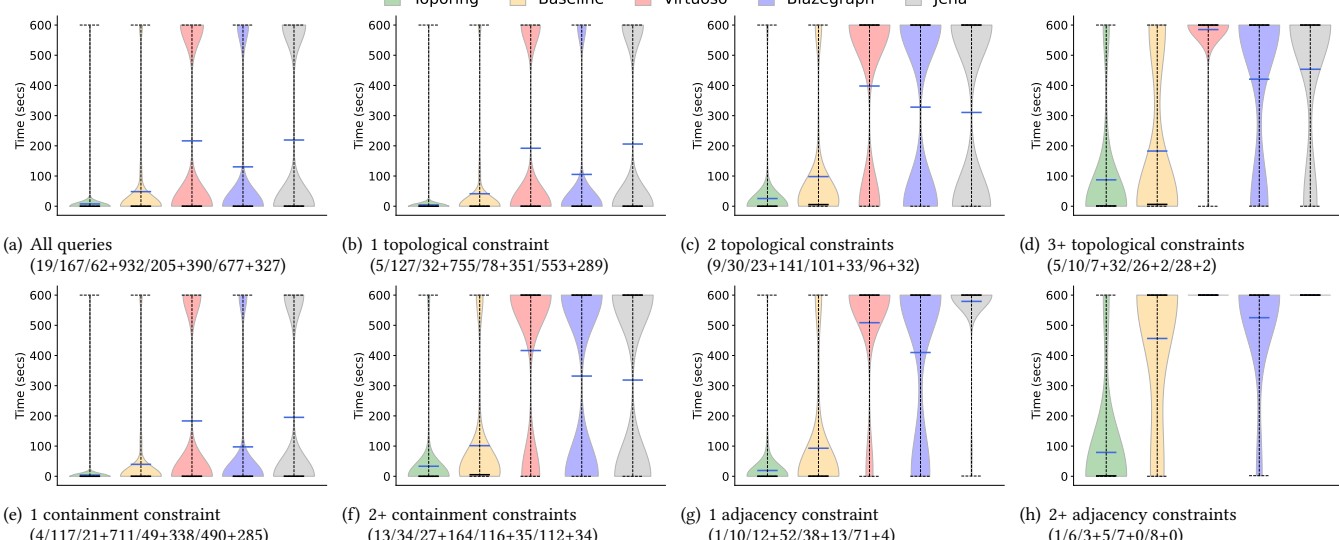

**Figure 4: Runtimes grouped by number and kind of topological constraints. For each system the number of queries with timeout is shown in parentheses, as (TopoRing/Baseline/Virtuoso timeouts+errors/Blazegraph timeouts+errors/Jena timeouts+errors).**

*Extended BGPs.* Figure 4 shows how the running times distribute on the tested queries. The overall results in Figure 4(a) show that TopoRing is significantly faster than the other alternatives, with an average query time of 7.52 seconds, 6.5 times less than that of our Baseline and an order of magnitude less than the average of Virtuoso, Blazegraph and Jena. While many queries are solved fast, as witnessed by the low medians, a significant part of them are indeed difficult. In particular, TopoRing times out on 19 queries, the Baseline on 167 queries, Virtuoso on 62, Blazegraph on 205, and Jena on 677 queries. However, we found that Virtuoso, Blazegraph and Jena also exhibit 932, 390 and 327 non-timeout errors, resp., due to an apparent bug processing zero-or-many (*) paths between two variables, where it throws an error or returns no results when results are expected. Adding up timeouts and errors, Virtuoso is unable to properly handle 994 queries (36%), Blazegraph 595 queries (21%) and Jena 1,004 queries (36%).

Figures 4(b)-4(d) separate the queries by the number of topological constraints. It shows that those constraints affect query times by a significant margin, driving the average times of TopoRing from 4.46 seconds with one constraint to 87.3 with three. The impact on the Baseline is even higher, and Virtuoso, Blazegraph and Jena solve very few queries with three constraints in less than 600 seconds.

Figures 4(e)-4(h) classify the queries according to the number of containment and adjacency constraints, respectively. It can be seen that the latter pose a heavier load to the TopoRing than the former, but again, the effect on the Baseline, Virtuoso, Blazegraph and Jena is much higher, up to the point that the median times out in both.

*Space usage.* Our TopoRing uses 15.5GB, that is, 17.38 bytes per triple (bpt). From this space, 12.30 bpt are used by the underlying Ring and 5.08 bpt by our structures handling topological constraints.

Virtuoso, Blazegraph and Jena, on the other hand, use 60.07, 90.79 and 95.83 bpt, respectively. This space includes the dictionary mapping between nodes and their strings. This mapping can be added to the Ring at a space cost of 3.68 bpt [4] with no impact on query time. With this mapping, our TopoRing would use 21.06 bpt.

## 9 CONCLUSIONS AND FUTURE WORK

Many knowledge graphs contain topological relations, often – but not exclusively – to represent geospatial relations, such as contains and touches. Herein we have proposed techniques that are efficient – in both time and space – for querying knowledge graphs, returning results entailed via the semantics of topological relations. We have formally characterized the efficiency of our approach, showing, for example, that it constitutes a worst-case-optimal algorithm. In practice, our approach provides notable speed-ups when compared with internal and external (SPARQL) baselines for evaluating a real-world workload of queries extracted from Wikidata logs.

In terms of limitations, our index structure does not currently permit updates; these could be supported via dynamic compact data structures for ordinal trees and wavelet matrices [35, 42]. Furthermore, the index structures we use work in RAM, where adapting them to work efficiently on the disk is non-trivial due to the random access patterns that they generate. Finally, our representation permits having hierarchies other than contains, as long as each element belongs to only one hierarchy. However, our model can be extended to allow for overlaps in hierarchies (see Appendix C).

The semantic properties of topological relations – in particular, symmetry and transitivity – may also apply to a broader class of relations. Indeed, through our SPARQL baseline, we showed that such semantics can be supported via (2)RPQs. However, for RPQs, it is unlikely that useful worst-case-optimal guarantees exist [12]. This raises the question of where, precisely, is the barrier for wco guarantees, and for what kinds of RPQs – or semantic properties – can algorithms boasting such guarantees be provided.

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

## A   PROOFS AND MISSING DETAILS

### A.1   Details of Theorem 1

The total space incurred by our data structures is $3N+n+(1+\epsilon)M = O(N+n)$ words of space (each of $\log n$ bits), plus sublinear terms, where $M \le N$ is the number of triples of the form $(x, \text{touches}, y)$. Precisely, we use:

- $3N \log n + o(N \log n)$ bits for a Ring built on the graph edges.
- $n \log n + O(n)$ bits to store a permutation between actual node identifiers and convenient internal identifiers.
- $M \log n + o(M \log n) + O(n)$ bits to store the adjacencies.
- $5n + o(n)$ bits to represent the hierarches and cases with zero bound variables.
- $(\epsilon M) \log n$ bits to support not-touches.

Combined with the worst-case optimality we prove in Section 7, we obtain the theorem.

### A.2   Proof of Lemma 1

PROOF.   By definition $y$ is an ancestor of $x$ and of $z$, so $postorder(y) \ge postorder(z) > postorder(x)$, thus $y \ne x$. We now prove that $y$ is the lowest node in the path to $x$ with large enough postorder, that is, its child $w$ towards $x$ has $postorder(w) < postorder(z)$.

Note that $z$ cannot descend from $w$ by definition of $lca$, so $z$ descends from a child $w' \ne w$ of $y$. Then $w$ must precede $w'$: otherwise, since $x$ descends from $w$, we would have $postorder(z) < postorder(x)$. Therefore, it holds that $postorder(w) < postorder(z)$, a contradiction.   □

### A.3   Containment and disjointness with zero or two bound variables

Checking any of the relations of Section 5 between two constants $x$ and $y$ is trivially carried out by computing $x \sqsubseteq y$ and $y \sqsubseteq x$. For two variables, the simplest implementation for the general functions contains($c$) and contained($c$) is to store a bitvector of length $n$ for each, using $succ_1(c)$ to find the next postorder that contains some node or is contained in some node. This yields constant time with just $2n+o(n)$ extra bits; recall Section 3.5. Functions not-contains($c$) and not-contained($c$) are implemented with $succ_0(c)$ on the same bitvectors. The corresponding functions for disjointness are trivial: disjoint($c$) is always $\bot$ if $\sqsubseteq$ forms a single tree consisting of a single path, otherwise it is always $c$. For not-disjoint($c$), we enumerate first the $t$ trees of $\sqsubseteq$ formed by isolated nodes, and thus not-disjoint($c$) is $t+1$ if $c \le t$, or else $c$.

### A.4   Proof of Lemma 2

PROOF.   Since $w = lca(x, z)$ is an ancestor of both $x$ and $z$, it holds that $first(w) \le postorder(x) < postorder(y) < postorder(z) \le postorder(w)$, and thus $y$ descends from $w$. Since both $y$ and $z$ descend from $w$, $w' = lca(y, z)$ descends from $w$ as well, and thus $postorder(w) \ge postorder(w')$.   □

### A.5   Adjacency with zero or two bound variables

The case of bound $x$ and $y$ can be checked as touches($x, postorder(y)$) = $postorder(y)$, in $O(\log n)$ time. For two bound variables, we again implement touches($c$) with a bitvector telling which postorders touch some other node, and answer it with $succ_1$ on the bitvector, in constant time. The negated adjacency, not-touches with zero or two bound variables, is handled analogously with no further space.

### A.6   Proof of Proposition 2

PROOF.   Let us begin with some terminology. We say that $Q$ is *irreducible* if there exists no set of variables or constants that form a cycle of containment constraints $x \sqsubseteq y$. Queries that are not irreducible can be made so by identifying cycles in containment constraints and replacing them with identities, as $x \sqsubseteq y, y \sqsubseteq x$ is equivalent to $x = y$. This is why we shall only focus on irreducible queries.

Further, let us first assume our query only uses constraints $u \sqsubseteq v$, $u \,|\, v$, or $u \,\sqcap\, v$. This proof is of independent interest since it provides bounds even under the assumption that regions must always be connected. We later explain how to extend this proof for the remaining operators.

Assume $Q$ involves $\ell$ triples and $k - \ell$ constraints, so that it has the form $\bigwedge_{i=1}^{\ell} T(y_i, z_i, w_i) \wedge \bigwedge_{i=\ell+1}^{k} C_i(u_i, v_i)$, where each $C_i(u_i, v_i)$ is one of $u_i \sqsubseteq v_i$, $u_i \,|\, v_i$ or $u_i \,\sqcap\, v_i$. Then $flat(Q)$ has the form $\bigwedge_{i=1}^{\ell} T_i(y_i, z_i, w_i) \wedge \bigwedge_{i=\ell+1}^{k} D_i(u_i) \wedge D_i(v_i)$.

Let $\overline{x} = vars(Q)$. We use the dual program of the AGM bound of $flat(Q)$, considering arbitrary integers $N$ and $n$ for the number of triples and nodes of a graph:

$$\text{maximize: } \sum_{x \in \overline{x}} v_x$$
$$\text{subject to: } v_{y_i} + v_{z_i} + v_{w_i} \le \log N, \quad i = 1, \dots, \ell$$
$$v_{p_i} \le \log n, \quad i = \ell + 1, \dots, k$$
$$v_{q_i} \le \log n, \quad i = \ell + 1, \dots, k$$
$$v_x \ge 0, \quad x \in \overline{x}$$

By duality, any solution $\sum_{x \in \overline{x}} v_x$ for the dual is always smaller than the corresponding primal solution, with equality when the solutions are optimal. Let us assume that $N$ and $n$ are of the form $2^{L_N}$ and $2^{L_n}$ for some $L_N, L_n \in \mathbb{N}$, so the optimal solution of both the primal and dual are rational. Let $(v_x)_{x \in \overline{x}}$ be the dual solution and write each $v_x$ as $p_x/b$. Then $(p_x)_{x \in \overline{x}}$ is an optimal solution to the linear program with cardinalities $N^b, n^b$. Now we present a graph $G$ with $\ell N^b$ triples and $\ell n^b$ nodes such that $|Q(G)| \ge 2^{p^*(flat(Q), (N^b, n^b))}$.

Vertices and triples of $G$ are as follows:

- The vertices of $G$ are the union of sets $V_x = \{a_1^x, \dots, a_{2p_x}^x\}$ for each $x \in \overline{x}$.
- For every triple $T_i(y_i, z_i, w_i)$ in $Q$, add to $G$ all tuples in $V_{y_i} \times V_{z_i} \times V_{w_i}$.

From the construction we verify that every triple $T_i(y_i, z_i, w_i)$ contributes with at most $2^{p_{y_i}+p_{z_i}+p_{w_i}} \leq 2^{b \log N} = N^b$ triples and every constraint $C(u_i, v_i)$ contributes $2^{p_{u_i}} + 2^{p_{v_i}} \leq 2 \cdot 2^{b \log n} = 2n^b$ additional nodes. All of this guarantees that the graph contains at most $\ell T^b$ triples and $\ell n^b$ nodes.

We next show how to construct the topology of the graph, for which we need a few more definitions. For a consistent, irreducible query $Q$, its *constraint graph* has an undirected edge from node $x$ to node $y$ for each constraint between $x$ and $y$ in $Q$. The *subset-constraint graph* has a directed edge from node $x$ to node $y$ for each constraint $x \sqsubseteq y$ in $Q$.

LEMMA 3. *If $Q$ is consistent and irreducible, then: (1) Its subset-constraint graph is acyclic. (2) If $y$ is reachable from $x$ in the subset-constraint graph of $Q$, then $Q$ cannot mention $x$ and $y$ together in any other constraint.*

PROOF. First item follows because cycles in the subset-constraint graph can be reduced, as these imply equality between all variables in the cycle. Second item follows because our hierarchy assumption implies that $x \sqsubseteq y$ is mandated by $Q$, and then if $x$ and $y$ are mentioned in any other constraint $Q$ would not be consistent. □

To construct the topology of $G$, consider the constraint-graph of $Q$. We make us of a result by Arseneva et al. [6], which states that for the constraint graph of $Q$ we can build a set of (interior) disjoint polygons in 3D, one for each node of the constraint graph, so that two polygons share a side if and only if the corresponding vertices are adjacent in the constraint graph. Note that this construction results in a set of polygons, one per each variable of $Q$, that share a side if and only if the query $Q$ contains a constraint (be it subset, adjacency or disjointedness) that mentions both variables.

Denote this set of polygons as $\Pi$. From $\Pi$ we construct the topology of $G$ via a series of refinements.

First, for each variable $x$ in the constraint graph of $Q$, recall we defined $V_x = \{a_1^x, \ldots, a_{2^{p_x}}^x\}$, and let $n_x = |V_x|$. Let $\pi^x$ be the polygon associated with variable $x$ in $\Pi$. Partition the polygon into $n_x$ subpolygons $\pi_1^x, \ldots, \pi_{n_x}^x$ in such a way that $\pi_1^x$ shares all the facets of $\pi_x$, such as in Figure 5. Then, associate each element $a_i^x \in V_x$ with the polygon given by $\bigcup_{j \leq i} \pi_j^x$. Denote this polygon as $p(a_i^x)$, and notice in particular that $p(a_{n_x}^x) = \pi^x$.

This construction ensures the following:

- For each variable $x$, all polygons associated with elements in $V_x$ satisfy all adjacency axioms previously satisfied by $\pi^x$. As a consequence, for every constraint $x \mid x'$ in $Q$, every element in $V_x$ is adjacent to every element in $V_{x'}$.
- Since adjacent polygons are considered disjoint, for each constraint $x \not\mid x'$ in $Q$ we have that every element in $V_x$ is disjoint to every element in $V_{x'}$.

We will further refine this topology to account for subset constraints in $Q$. We deal separately with each connected component of the subset-constraint graph of $Q$. For each such component $C$, which by Lemma 3 is a DAG, we first ensure that every node in $C$ has at most one outgoing edge, by iteratively replacing edges $(x, y)$ and $(x, z)$, corresponding to constraints $x \sqsubseteq y$

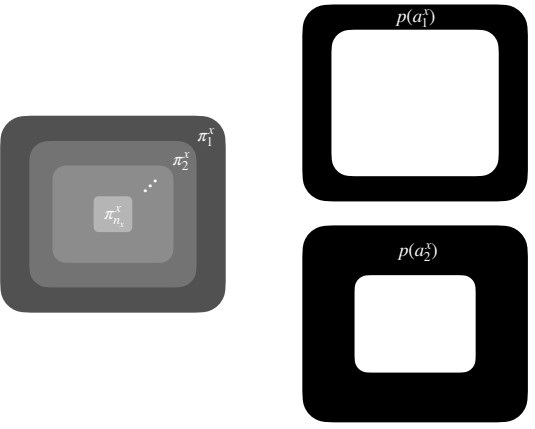

Figure 5: Partition of polygon $\pi^x$. Each node $a_i^x$ in the graph is then associated with polygon $p(a_i^x) = \bigcup_{j \leq i} \pi_j^x$.

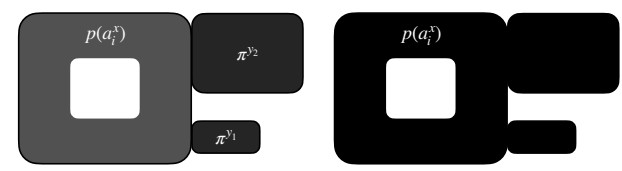

Figure 6: Redefinition of $p(a_i^x)$ when facing constraints $y_1 \sqsubseteq x$ and $y_2 \sqsubseteq x$ from the initial construction (left) to the final result (right).

and $x \sqsubseteq z$, with $(x, y)$, $(y, z)$.[6] Importantly, this modification ensures that for every pair $x, z$ of nodes in the component $C$ of the constraint graph, if $x$ is an ancestor of $z$ in $C$, then $x$ is also an ancestor of $z$ in the modified graph. Pick one node traversal of this graph. In this order, we do the following. If a variable $x$ has no ancestor, that is, if there is no $y$ such that $(y, x)$ is in $C$, we leave all $n_x$ polygons for $x$ as constructed before. Otherwise let $y_1, \ldots, y_k$ be the nodes such that edges $(y_1, x), \ldots, (y_k, x)$ belong to the subset-constraint graph. Then, for each element $a_i^x$ in $V_x$, redefine $p(a_i^x) = p(a_i^x) \cup \pi^{y_1} \cdots \cup \pi^{y_k}$. Figure 6 depicts this construction for a specific polygon $p(a_i^x)$ that shares a side with two other polygons $\pi^{y_1}$ and $\pi^{y_2}$.

Notice, then, that each of the polygons associated with elements in $V_{y_1}, \ldots, V_{y_k}$ are contained in all the polygons associated with any element in $V_x$. Hence, the topology so far satisfies the subset constraints in the modified graph, and since the modification preserves ancestry, it also satisfies the subset constraints in $C$. Moreover, for each constraint $y_j \sqsubseteq x$ we have that every element in $V_{y_j}$ is contained in any element in $V_x$ Furthermore, all other constraints in $Q$ mentioning variable $x$ continue to be satisfied, as the only disjoint/adjacent constraints that are falsified in this construction involve two variables in $C$ where one is an ancestor of the other, which we assume not to exist by Lemma 3.

---

[6]Since we assume topologies are hierarchies, whenever $a \sqsubseteq b, a \sqsubseteq c$ hold we have either $a \sqsubseteq b, b \sqsubseteq c$ or $a \sqsubseteq c, c \sqsubseteq b$. Hence, with this refining we are forcing one of these two cases for every element realizing the original constraints.

Concerning the evaluation, since the graph contains all combinations of the corresponding nodes participating in one of the triples in $Q$, and since constraints $C(x, y)$ in $Q$ are realized in all elements in $V_x \times V_y$, we immediately obtain that that evaluation $Q(G)$ contains all tuples $t \in V_{x_1} \times \cdots \times V_{x_n}$. We now have a graph $G$ with the desired cardinality profile for which

$$Q(G) \geq 2^{\sum_{x \in \overline{x}} p_x};$$

the right item corresponds to $2^{\rho^*(flat(Q),(N^b,n^b))}$ by duality.   □

*Extending the proof for negated operators.* To deal with the remaining operators, we need to modify the construction above so that polygons for constraints $x \not\boxminus y$ in the constructed graph are not adjacent but disconnected. We can further show that this modified construction continues to serve as a lower bound for queries even if we add an additional constraint that enforces that two regions are disconnected. Finally, to show the bound for our full query language, all we need to do is to use our axioms to rewrite queries so that they only use adjacency, subset, disjoint and disconnected constraints, for which we can apply the result outlined in the previous paragraph. Details are omitted due to space limitations.

Our proof provides lower bounds even in a restricted topological space like polygons in $\mathbb{R}^3$. We can even build a worst-case instance in $\mathbb{R}^2$ if topological regions may correspond to a set of disconnected areas in the plane: for every $C(x, x')$ we create two touching areas disconnected from all the others, one belonging to the region of $x$ and the other of $x'$. We can further enforce that regions are simple connected areas if the constraints graph of the query is planar. In such a case, a planar embedding of its dual produces a set of regions, one per variable, that touch each other iff they are connected by a constraint. Those regions play the role of the polygons of Arseneva et al. [6].

## B DETAILS ON THE EXPERIMENTAL RESULTS

*Variable elimination order in TopoRing.* It is well known that the order in which variables are eliminated may have a huge impact in practice on the running times of LTJ [54]. The Ring chooses the order based on estimating the number of solutions to triple patterns. We extend such estimations to topological relations using our data structures. We estimate the cardinality $c(t)$ of a topological constraint $t$ as follows: If no variable is bound, $c(t)$ is estimated as the total number of nodes with that topological relation. If $t$ is of the form $x \sqsubseteq y$ and $x$ is bound, then $c(t)$ is estimated as the depth of the node to which $x$ is bound. If instead $y$ is bound, then $c(t)$ is estimated to be the number of descendants of the node to which $y$ is bound. Those values are computed in constant time with the data structure for ordinal trees described in Section 3.5. If $t$ is of the form $x \mid y$ and one of $x$ and $y$ is bound, then we (under)estimate $c(t)$ as the number of adjacency axioms of the descendants of the bound variable $x$.

*Dataset and queries.* We extracted all queries containing a single BGP, filtering those not mentioning topological properties, any disconnected BGPs that would invoke a Cartesian product, and BGPs that are duplicate with respect to isomorphism of variables.

We selected two predicates from Wikidata to represent the containment relation: P150 (*contains the administrative territorial entity*)

and P131 (*located in the administrative territorial entity*), and the predicate P47 (*shares boundary with*) to represent the adjacency relation. The data structures presented in Section 5 need to have a forest on the relation $\sqsubseteq$, but this dataset is, in fact, a directed acyclic graph. To overcome this problem, we retained only a spanning tree of the directed acyclic graph by deleting 1,393,677 triples. We further include edges for all other predicates in the graph, but without any special interpretation. The complete resulting dataset has $N = 957,450,487$ triples (107,836,911 subjects, 242,124,917 objects, 5,419 predicates, and $n = 296,008,192$ unique nodes). From those triples, 6,881,975 correspond to containment and 499,741 to adjacency. The selected query log contains 2,782 queries, each of which mentions at least one of the predicates P150, P131, or P47.

*Translating queries to SPARQL.* A topological constraint $x \sqsubseteq y$ is translated into the following:

```
SELECT DISTINCT ?x ?y {
    ?x (P150|^P131)* ?y .
    ?x (P150|^P150|P131|^P131|P47|^P47) ?x_e . }
```

With the auxiliary variable $x\_e$, we make sure that the query only binds $x$ to a node that participates in topological relations. A constraint of the form $x \mid y$ is translated into the following syntax:

```
SELECT DISTINCT ?x ?y {
    ?x (P150|^P131)* ?x_s .
    ?y (P150|^P131)* ?y_s .
    ?x_s (P47|^P47) ?y_s .
    FILTER NOT EXISTS {
        ?x ((P150|^P131)*|(^P150|P131)*) ?y } }
```

This extended query captures the fact that $x$ or a sub-region of $x$ touches $y$ or a sub-region of $y$, and the filter excludes the cases where $x$ contains $y$ or vice-versa.

*Experimental setup.* Experiments were run on a machine with two Intel Xeon Silver (4316) processors, clocked at 2.30GHz; 251GB RAM memory clocked at 3,200 MT/s; 40 physical cores each one with L1i, L1d and L2 caches of size 32KB, 48KB and 1,280KB, respectively; and a L3 cache of size 30MB. The machine runs Linux 5.14.0-162.22.2.el9_1, in 64-bit mode. The code was compiled with GCC 11.3.1 using flags -msse4.2 -O3 -ffast-math -funroll-loops -fno-omit-frame-pointer. TopoRing times are averaged over 4 executions. Because they were much slower, Baseline times were averaged over 2 executions and Virtuoso, Blazegraph and Jena over 1 execution. A timeout of 10 minutes was set per query.

## C RELATING MULTIPLE HIERARCHIES

Our representation permits having the elements distributed across more than one hierarchy, though each element must belong to only one. This is represented by a set of Hasse diagrams, or as explained, as a forest of hierarchy trees, and concretely as a concatenation of their parenthesis sequences. By definition, there cannot be containment relations between different hierarchies (in particular, an object cannot be contained in two objects that are not one contained in the other). We do support having adjacency relations between different hierarchies, which may fit some applications.

*Allowing overlaps.* In other cases, there may be *overlap* between elements from different hierarchies (e.g., two hierarchies of administrative subdivisions). This information will be considered when answering whether two regions are disjoint: we said until now that $x \sqcap y \Leftrightarrow x \sqsubseteq y \vee y \sqsubseteq x$. We now add another possibility for $x \sqcap y$ to hold: $x$ and $y$ are declared to overlap in an axiom (i.e., graph triple) $(x, \text{overlaps}, y)$. Not-disjointness will then be determined by, in addition to containment, the overlapping axioms stored in a binary matrix $O[i, j]$, and enforcing a rule analogous to a2:

o1. If $x \sqcap y$, $x \sqsubseteq x'$, and $y \sqsubseteq y'$, then $x' \sqcap y'$.

Our extended algorithm to answer not-disjoint$(x, c)$ takes the smallest between two candidates. The first is obtained exactly as before (we return $first(x)$ if $c < first(x)$; else we return $c$ if $node(c) \sqsubseteq x$; else we return contained$(x, c)$). The second is obtained from the matrix $O$ much as with touches, yet in simplified form because rule o1 is simpler than a2 (i.e., it allows returning nodes containing $x$): among the cells $O[i][j] = 1$ for $first(x) \leq i \leq postorder(x)$, we choose the minimum among the smallest $j \geq c$, and $lca(node(j), node(c))$ for the largest $j < c$. Recall that, in the actual algorithm, we perform four searches on matrix $O$ instead of two, to avoid storing each axiom twice. Overall, this takes time $O(\log n)$ and stores the overlap axioms only once.

The case for disjoint$(x, c)$ is analogous: we take the smallest between two candidates. The first is obtained exactly as before (we return $c$ if $c < first(x)$; else, if $c \leq postorder(x)$, we reset $c \leftarrow postorder(x) + 1$; and in either case return not-contained$(x, c)$). The second candidate is obtained from the matrix $O$ much as with not-touches, for which we showed how to answer queries in time $O((h/\epsilon) \log n)$ using an $O(\epsilon)$ fraction of extra space.

*Independent hierarchies.* If we do need objects to belong to different hierarchies (as in our original Wikidata graph), we can create one distinct identifier $id_h$ per hierarchy $h$, and add a triple $(id_h, \text{corresponds\_to}, id)$, where $id$ is the global identifier of the object. Those triples must be considered in queries, as we should relate only the global identifier to other nodes. All of these extensions continue to support LTJ's *leap* operation in the required time. More precisely, since there is only one hierarchy per object, the triples $(id_h, \text{corresponds\_to}, id)$ involve a *key* constraint. To obtain the AGM bound for these types of queries one needs to *chase* these key constraints, as done by Gottlob et al. [23], but LTJ is still optimal in this case with respect to the chased query. Hence, our algorithm maintains worst-case optimality, as stated in Theorem 1.

