# OpenReview forum: "Worst-Case-Optimal Joins on Graphs with Topological Relations"
_ACM.org/TheWebConf/2025/Conference — WWW 2025 Oral_

### Official Review · Reviewer_nAT9 · 2024-11-24

**Novelty:** 5
**Technical Quality:** 6

**Review:**

Advantages:
1. Proposes an efficient query processing method for topological relationships, specifically extending the Leapfrog Triejoin (LTJ) algorithm, demonstrating significant advantages in query performance and space efficiency.
2. Provides detailed experimental results, showcasing improvements in time and space complexity, with superior performance compared to other methods (e.g., SPARQL engines) on multiple benchmarks.
3. Offers clear definitions and modeling of topological relationships and constraints, aiding readers in understanding the semantics of complex queries.

Weaknesses:
1. The experiments focus on small-scale graph query patterns, which may not fully reflect the complexity of large-scale datasets in real-world production environments.
2. The comparative analysis is primarily limited to specific benchmarks, with a lack of discussion on other modern query optimization methods.

Comments:
1. The theoretical derivations are clear and easy to follow, but certain technical details, such as the index structure and parameter tuning, could benefit from further elaboration.
2. The experimental results effectively demonstrate the advantages of the method, but research on its adaptability to other domains of databases could enhance its applicability.
3. While the paper focuses on low-level algorithm optimization, it lacks a detailed discussion on how users define and utilize topological constraint queries (e.g., through user interfaces or query language extensions). Including this aspect could improve its engineering value.

**Questions:**

1.The submission guidelines explicitly require clarification of the paper's relationship with the Web. How does your research relate to the Web?

2. Is using sentence similarity as the sole evaluation metric scientifically sufficient? Are there other metrics that could be included?

3. What scenarios can benefit from your scientific problem? Are there related tasks that could gain from your research?

**Reviewer Confidence:**

2: The reviewer is willing to defend the evaluation, but it is likely that the reviewer did not understand parts of the paper

**Scope:**

2: The connection to the Web is incidental, e.g., use of Web data or API

---

### Official Review · Reviewer_DM3r · 2024-11-25

**Novelty:** 5
**Technical Quality:** 6

**Review:**

This paper describes an extension to the Leapfrog Triejoin algorithm so that it can be used to answer queries over topological relationships in a knowledge graph (relations such as containment, disjointness, touching etc).  This paper is outside my space of expertise so I defer to someone else who might know a lot more about the space.

**Questions:**

1.  What is the rationale of querying topological relations in this way?  As you point out, specialized GIS systems have been around for a long time, and they are used heavily.  Why would one want to combine knowledge graphs and topological relations this way?  Isn't it more natural to express locations with geometric points rather than these higher level abstractions?  Don't the GIS systems already allow you to ask higher level relations such as touching?  I did see your examples but they were not especially compelling.
2.  I might be wrong but I got the impression that your approach codifies the querying of these topological relations in some mechanism other than SPARQL because as you point out, expressing this in SPARQL requires as you put it well  -"resorting to more complex queries that combine BGPs, RPQs and negation."  If I surmised that correctly and you did not use SPARQL - then we go back to my question 1 - why would anyone want to represent and query topological relations in this way instead of using a specialized GIS system.
3.  If the goal was to combine KG triples with topological relations is it possible to think of a separate index that handles GIS data, and think of a federating a query over those two sources, analogous to how text indexes and text queries can be used with KG triples - see Allegrograph as an example: (https://franz.com/agraph/support/documentation/4.7/sparql-geo.html).
4.  The graph used in the experiments was the entire Wikidata?  That seems too large to store the graph in memory and much of it is likely irrelevant for spatial queries?  Can you say a bit more about the graph as well as the 'graph patterns' you extracted?  What was the constraints used for gathering those?

**Reviewer Confidence:**

1: The reviewer's evaluation is an educated guess

**Scope:**

3: The work is somewhat relevant to the Web and to the track, and is of narrow interest to a sub-community

---

### Official Review · Reviewer_se8L · 2024-11-29

**Novelty:** 6
**Technical Quality:** 6

**Review:**

The work addresses qualitative spatial queries, i.e. queries concerning containment or connection of spatial reasons rather then quantitative queries concerning for instance distances and areas.  This is an important field since many queries in a geospatial setting are in fact qualitative, and methods meant for quantitative queries and data are an overkill.

The submission presents specialised compact data structures and algorithms that allow worst-case optimal execution of such queries.  They are presented in detail and their properties are proven.

I am not an expert but expect the work to be interesting within the field of spatial knowledge graphs.  I have a few issues with the presentation however.

1. There is a difference between queries and data: it is quite possible to optimise qualitative queries over existing quantitative data, see e.g. https://link.springer.com/article/10.1007/s10707-019-00348-z  As soon as quantitative data is available, reasoning is far less important, since each containment or adjacency can checked directly.  It remains to store information in a compact manner.

In the submission, I found it a bit difficult to find out what was actually assumed about the dataset that was being queried, and I would ask the authors to clarify this.  I _think_ this work is based on purely qualitative data to begin with (which is fine if made explicit).  Much of the work referred to in the related work section concerns quantitative data and would be worth pointing that out.

2. The paper makes a rather important assumption in Section 3.2, namely that the containment relation forms a mono-hierarchy: if two spatial entities overlap, then one of them has to contain the other.  This restriction is crucial for the data structures proposed.  There is no discussion of how realistic this assumption is, what can be done if it is not met (you could subdivide regions, but what does that do to complexity?)  But more seriously the fact that there is this kind of restriction compared to most of the related work is somewhat hidden in the technical discussion.  It would be better to announce very early in the text that this paper solves the problem "for the important subclass of hierarchical spatial knowledge bases" or something like that.

3. Even later, in section 4.1, the authors announce that they use a negation as failure semantics.  I’m not sure what the consequences of this choice are.  If it doesn’t really matter, please explain why.  If it does matter (i.e. negation in the qualitative queries doesn’t really mean negation) the this should again be stated more prominently.

**Questions:**

In a large knowledge graph, e.g. stemming from data integration, we can expect some spatial entities to be given with their (quantitative) geometry, while only topological information is given for others.  How would you envision to extend your approach for such a mixed scenario?

How would you method extend to spatial-temporal information?

Are these data structures and algorithms good for any kind of data taken from a hierarchy of possible values?  E.g. type hierarchies?  Can this be generalised to entities belonging to several hierarchies, like a type hierarchy as well as a part-of hierarchy?

**Reviewer Confidence:**

2: The reviewer is willing to defend the evaluation, but it is likely that the reviewer did not understand parts of the paper

**Scope:**

3: The work is somewhat relevant to the Web and to the track, and is of narrow interest to a sub-community

---

### Official Review · Reviewer_dpCy · 2024-11-30

**Novelty:** 6
**Technical Quality:** 6

**Review:**

The author(s) tackled the problem of incorporating topological information into knowledge graph queries by designing a compact index based on the Hasse diagram of containment and disjointness relations with balanced parenthesis, enabling constant-time operations. They extended the Leapfrog Triejoin (LTJ) algorithm to efficiently handle these topological constraints, ensuring the leap operation operates within O(log*n) time. Their approach was tested on the truthy Wikidata graph against a non-trivial baseline they implemented with Ring and the data structures in the paper, Virtuoso, Blazegraph, and Jena.


Strengths
1. Introduced a novel worst-case-optimal algorithm for evaluating basic graph patterns with topological relations, achieving both space and time efficiency.
2. Clearly explains the underlying core concepts of their topological model with binary relations data structures such as Bitvectors, Permutations, Ordinal Trees, and Wavelet matrices. There is a clear explanation of the data structure’s handling of containment and disjointness.
3. The paper identifies related work within spatial databases and topology in graph databases from other academic papers.
4. The paper clearly explains the variant Leapfrog Triejoin algorithm for basic graph patterns and how they extend the Ring Leapfrog Triejoin implementation to fit their constraints while maintaining the worst-case-optimal.
5. There are figures within the paper that provide a clear visual representation of what the author(s) are presenting.
6. There is a clear outline of limitations with their index structure, such as not permitting updates and being in RAM. They also mention other extensions people can implement and acknowledge how this would be achieved within the Appendix.
7. The paper provides important context and missing information within the appendix from their theorems and proofs alongside their rules. Additionally, the Appendix includes more details of their experiments.

Weaknesses
1. In section 3.2, line 273, the “x | y: not” relation is incorrect as the “not adjacent” should have a slash through the symbol.
2. More information and explanation on the usage of SPARQL would be beneficial.

**Questions:**

See the weakness section.

**Reviewer Confidence:**

1: The reviewer's evaluation is an educated guess

**Scope:**

4: The work is relevant to the Web and to the track, and is of broad interest to the community